# Utility of Clustering in Mortality Risk Stratification in Pulmonary Hypertension

**DOI:** 10.3390/bioengineering12040408

**Published:** 2025-04-11

**Authors:** Pasquale Tondo, Lucia Tricarico, Giuseppe Galgano, Maria Pia C. Varlese, Daphne Aruanno, Crescenzio Gallo, Giulia Scioscia, Natale D. Brunetti, Michele Correale, Donato Lacedonia

**Affiliations:** 1Department of Medical and Surgical Sciences, University of Foggia, 71122 Foggia, Italy; pasquale.tondo@unifg.it (P.T.); mariapiavarlese@gmail.com (M.P.C.V.); giulia.scoscia@unifg.it (G.S.); natale.brunetti@unifg.it (N.D.B.); donato.lacedonia@unifg.it (D.L.); 2Respiratory and Intensive Care Unit, Policlinico Foggia, 71122 Foggia, Italy; daphne.aruanno@unifg.it; 3Cardiothoracic Department, Policlinico Foggia, 71122 Foggia, Italy; lucia.tricarico.lt@gmail.com (L.T.); opsfco@tin.it (M.C.); 4Department of Cardiology, F. Miulli Hospital, Acquaviva delle Fonti, 70021 Bari, Italy; giuseppegalgano@hotmail.com; 5Department of Clinical and Experimental Medicine, University of Foggia, 71122 Foggia, Italy

**Keywords:** clustering, machine learning, predictive mortality factors, pulmonary hypertension

## Abstract

**Background**: Pulmonary hypertension (PH) is a condition characterized by increased pressure in the pulmonary arteries with poor prognosis and, therefore, an optimal management is necessary. The study’s aim was to search for PH phenotypes and develop a predictive model of five-year mortality using machine learning (ML) algorithms. **Methods**: This multicenter study was conducted on 122 PH patients. Clinical and demographic data were collected and then used to identify phenotypes through clustering. Subsequently, a predictive model was performed by different ML algorithms. **Results:** Three PH clusters were identified: Cluster 1 (mean age 68.57 ± 10.54) includes 57% females, 69% from non-respiratory PH groups, and better cardiac (NYHA class 2.61 ± 0.84) and respiratory function (FEV1% 78.78 ± 21.54); Cluster 2 includes 50% females, mean age of 71.36 ± 8.32 years, 44% from PH group 3, worse respiratory function (FEV 1% 68.12 ± 10.20); intermediate cardiac function (NYHA class 3.18 ± 0.49) and significantly higher mortality (75%); Cluster 3 represents the youngest cluster (mean age 61.11 ± 13.50) with 65% males, 81% from non-respiratory PH groups, intermediate respiratory function (FEV1% 70.51 ± 17.91) and worse cardiac performance (NYHA class 3.22 ± 0.58). After testing ML models, logistic regression showed the best predictive performance (AUC = 0.835 and accuracy = 0.744) and identified three mortality-risk factors: age, NYHA class, and number of medications taken. **Conclusions**: The results suggest that the integration of ML into clinical practice can improve risk stratification to optimize treatment strategies and improve outcomes for PH patients.

## 1. Introduction

Pulmonary hypertension (PH) is a rare heterogeneous disease characterized by elevated blood pressure in the pulmonary circulation, caused by both pulmonary vascular remodeling and inflammation or increased downstream pressures [1]. This condition ultimately results in right heart failure and significantly contributes to morbidity and mortality among affected patients. The complexity of PH arises from its diverse etiologies, including idiopathic origins, connective tissue diseases, congenital heart defects, left heart disease, lung disease and chronic thromboembolic events, which are categorized into five distinct clinical groups by the World Health Organization (WHO) [1]: pulmonary arterial hypertension (PAH) (group 1), pulmonary hypertension due to left heart disease (PH-LHD) (group 2), pulmonary hypertension due to chronic lung disease (PH-CLD) and/or hypoxia (group 3), pulmonary hypertension associated with pulmonary artery obstructions (group 4), and pulmonary hypertension with unclear and/or multifactorial mechanisms (group 5).

The hemodynamic definition of PH is the elevation of mean pulmonary arterial pressure (mPAP) > 20 mmHg, assessed by right heart catheterization (RHC). Pulmonary arterial wedge pressure (PAWP) and pulmonary vascular resistance (PVR) distinguish pre-capillary PH (PAWP ≤ 15 mmHg, PVR > 2 Wood Units (WU), isolated post-capillary PH (PAWP > 15 mmHg, PVR ≤ 2 WU) and combined post- and pre-capillary PH (PAWP > 15 mmHg, PVR > 2 WU) [1,2].

PH affects all age groups, with an estimated prevalence of 1% of the world’s population. The prevalence is higher in patients aged > 65 years due to increased cardiac and pulmonary etiologies, with lung disease and chronic obstructive pulmonary disease (COPD) being a common cause of PH1.

Patients usually present with non-specific symptoms, such as dyspnea on exertion or at rest, fatigue and rapid exhaustion, bendopnea, angina, syncope and/or clinical signs of right heart failure [3].

Despite advancements in therapeutic options over the past decades, PH remains challenging to diagnose and manage due to its heterogeneous nature and the variability in disease progression among patients. Accurate and early diagnosis, coupled with effective management strategies, are crucial for improving patient outcomes and quality of life. However, underdiagnosis PH is a significant issue in clinical settings. Various factors impede the timely detection of this condition: diagnostic delay, due to the lack of specific symptoms and/or of awareness among general practitioners that can result in delayed referrals to specialized PH centers, limited access to diagnostic procedures, and variability in clinical practices and interpretation of results [4].

Currently, there are only two validated screening algorithms for pulmonary hypertension (PH): one for PAH secondary to scleroderma [5] and another for chronic thromboembolic pulmonary hypertension (CTEPH) following pulmonary embolism [6].

The prognosis of pulmonary hypertension is variable and depends on a mixture of hemodynamic, clinical, etio-pathologic parameters and serum biomarkers. Early diagnosis, accurate risk stratification, and appropriate treatment are essential to improve outcomes.

Specific treatment of PH depends on the disease group. Current therapies for pulmonary arterial hypertension (PAH, group 1) work through four major pathways: endothelin-1, nitric oxide, prostacyclin and bone morphogenetic protein/activin signaling. Oral, inhaled and parenteral medications are available. Their efficacy has generally been greater with therapeutic combinations and with parenteral therapy compared with monotherapy or non-parenteral therapies. The maximal medical therapy is the four-drug approach [7]. Reassessment and early escalation of therapy is recommended [8]. Sotatercept, a novel biologic agent targeting the transforming growth factor-β superfamily, acts as a ligand trap for activins and related growth factors [9,10].

The cornerstone of PH-LHD and PH-CLD (respectively, group 2 and group 3) therapy is management of the underlying clinical condition [1]. Regarding group 3, there is promising evidence of safety and efficacy for the use of inhaled treprostinil in patients with pulmonary hypertension associated with interstitial lung disease (PH-ILD) [11].

Pulmonary thromboendarterectomy (PTA) is the treatment of choice for CTEPH (group 4). Pulmonary vasodilators or balloon pulmonary angioplasty (BPA) are safe and effective in inoperable or post-PTA persistent/recurrent CTEPH. Riociguat is the only approved medical drug for CTEPH [12].

Lung transplantation remains an option for selected patients with an inadequate response to therapies [1,7].

The management of PH requires a multidisciplinary approach in specialized centers, with continuous assessment of treatment response and adjustment of therapy based on the risk profile.

Artificial intelligence (AI), particularly machine learning (ML), has shown promise in predicting cardiac conditions, including PAH [13,14]. Given that PAH patients tend to have high healthcare resource utilization before diagnosis [15], ML could potentially use this data for early detection [16]. Unsupervised machine learning approaches, such as clustering techniques, are increasingly being applied to complex, heterogeneous diseases, like PH, to identify distinct patient subgroups with similar clinical features or risk profiles. This data-driven stratification has the potential to transform real-life clinical practice by moving beyond traditional classification systems based solely on etiology (e.g., WHO groups) or hemodynamic parameters.

Clustering can reveal novel phenotypes of PH that may not be apparent through standard diagnostic criteria, thus enabling more personalized management strategies. For instance, studies have demonstrated that cluster-based stratification can uncover subgroups with significantly different prognoses or treatment responses, even within the same WHO group classification [17,18]. Such insights can inform treatment escalation decisions, monitoring strategies, and risk assessment beyond what current guidelines suggest.

From a clinical standpoint, cluster-informed risk models can guide tailored therapies—for example, identifying low-risk patients who might benefit from monotherapy versus high-risk clusters who may require early combination therapy or enrollment in clinical trials. Additionally, clustering can highlight the influence of non-hemodynamic factors, such as comorbidities, biomarkers, or exercise tolerance, offering a more holistic view of disease burden [19].

On a broader scale, incorporation of clustering approaches into future versions of clinical guidelines could support more individualized treatment pathways. This paradigm shift—from a one-size-fits-all approach to a precision medicine model—aligns with the growing emphasis on patient-centered care in pulmonary vascular medicine. However, for clustering to influence guidelines meaningfully, external validation, clinical interpretability, and prospective integration into decision-making tools will be essential.

In summary, cluster analysis holds promise not only for refining risk stratification and improving prognostic accuracy but also for reshaping clinical pathways and potentially influencing future guideline recommendations in the management of PH.

Along these lines, the present study aims to identify homogeneous patient clusters within a PH population and develop a predictive model for 5-year mortality, using ML techniques, thereby enhancing clinical decision-making and patient management.

## 2. Methods

### 2.1. Study Population

A prospective study was conducted on 142 patients evaluated for suspected PH by expert cardiologists and pneumologists from the University-Hospital Polyclinic of Foggia and the Miulli Hospital of Acquaviva delle Fonti between 2009 and 2018. PH diagnosis was confirmed through RHC, and patients were followed for five years. During baseline evaluation, we collected several forms of data including demographics (gender, age), clinical information (date of first diagnosis, PH group, NYHA classification, and number of medications), results from spirometry, DL_CO_, 6 Minute Walking Test (6MWT), RHC and, at 5-year from diagnosis, survival rate was assessed. The study was conducted according to Helsinki principles and approved by the Foggia Polyclinic EC (approval n. 177/C.E./2024). Patients signed for consent before being enrolled in the study

### 2.2. Cardio-Pulmonary Tests



*Spirometry*



Spirometry was performed using standardized equipment and technique, as defined by the American Thoracic Society/European Respiratory Society (ATS/ERS) task force [20]. All applicants performed three forced vital capacity (FVC) maneuvers, and the best of the 3 measurements was recorded. We collected the following spirometry data: FVC, forced expiratory volume in one second (FEV1), peak expiratory flow (PEF), Tiffeneau index (FEV1/FVC ratio), and forced expiratory flow during the middle half of the FVC maneuver (FEF25–75%). All spirometry values were corrected for height, weight, age, and sex.



*DL_CO_*



Diffusion within the lungs is an electrochemical process that occurs between the gas and liquid phases, driven by the gradient of partial pressures of the gases involved. The diffusion capacity quantifies the volume of any gas (mL) that passes through the alveolocapillary membrane per unit time (in minutes) under a specified pressure difference (1 mmHg). This capacity is expressed in units of mL/min/mmHg. The measurement is conducted with hemoglobin correction in accordance with the standards established by the European Respiratory Society and the American Thoracic Society (ERS/ATS) [21].



*Six Minute Walking Test*



Six minutes walking test (6MWT) was performed according to ATS 2002 guidelines for all patients as a marker of exercise tolerance with assessment of distance a patient can walk in six minutes and desaturation difference between the baseline SpO_2_ and post-exercise SpO_2_ [22].



*Right Heart Catheterization*



Hemodynamic assessment was conducted using a Swan–Ganz catheter (CCOmbo V, Edwards Lifesciences, Irvine, CA, USA). Measurements of systolic, diastolic, and mPAP, right atrial pressure (RAP), and pulmonary capillary wedge pressure were taken at the end of a quiet respiratory cycle. Oxygen saturations were recorded in the superior vena cava, inferior vena cava, pulmonary artery, and femoral artery, with pulmonary vein saturation assumed at 98%. Pulmonary and systemic flows were calculated using the Fick principle. Pulmonary and systemic vascular resistance indexes were determined using standard formulas. A pulmonary arterial wedge pressure greater than 15 mm Hg excluded the diagnosis of precapillary PAH.

### 2.3. Data Analysis

The statistical analysis was conducted using IBM SPSS Statistics version 26 (IBM Software, Armonk, NY, USA) [23]. The distribution of the sample was analyzed by Shapiro–Wilk test. Continuous variables were presented as mean and standard deviation (SD), while categorical variables were expressed as percentages. The data included in the analysis were used to identify homogeneous subgroups of patients (phenotypes) through clustering (Figure 1).

Differences between clusters were assessed using one-way ANOVA, with post hoc Tukey analysis validating the results. Mortality among the phenotypes was represented by Kaplan–Meier curves and analyzed using the log-rank test. A *p*-value of less than 0.05 was considered statistically significant.



*Unsupervised and supervised analysis*



ML analyses (clustering and predictive methods) were performed using the Orange software, version 3.38.1 [24].

Patients with confirmed diagnosis of PH were considered for analysis, and all collected variables were utilized. Through hierarchical clustering three homogeneous subgroups (clusters) were identified by means of Ward’s linkage method. The number of clusters was confirmed by the Silhouette plot index, calculated using Manhattan distance.

Subsequently, the Fast Correlation Based Filter (FCBF) ranking method was employed for searching the main subgroups features which were outcome-related. This ranking method identified three features that best determined the outcome of this study: the number of medications, age, and NYHA class.

After features selection, various ML algorithms were used to develop a model for predicting mortality in PH: Logistic Regression, Support Vector Machine (SVM), Random Forest, and Neural Network. The results of the models obtained from the four selected algorithms were validated using 10-Fold Cross Validation. The effectiveness of the models was evaluated using Area under the Receiver Operating Characteristic (ROC) curve (AUC) and Classification Accuracy (CA). Based on these metrics, Logistic Regression was found to be the most effective method (AUC = 0.835 and CA = 0.744) for the target variable (Figure 2), represented by 5-year survival from diagnosis.

## 3. Results

Out of 142 patients, 122 with confirmed PH were included in the final analysis. Population characteristics are listed in Table 1.

The study involved patients with a mean age of 67.13 ± 11.64 years, of which 52% were male and 48% were female. We divided the patients into two main groups based on the 2022 ESC/ERS guidelines: 31% to “Group 3” (PH associated with lung diseases and/or hypoxia) and 69% belonged to the “Other Group” (patients from groups 1, 2, 4, and 5 of the clinical classification of PH).

The average pulmonary function values showed an FVC of 77.56 ± 19.10% of the predicted value and an FEV1 of 73.12 ± 18.23% of the predicted value, with an FEV1/FVC ratio of 76.40 ± 13.20. The DL_CO_ was 48.20 ± 18.92% of the predicted value, while the DL_CO_/VA ratio was 74.18 ± 21.95. Hemodynamic parameters indicated a sPAP of 68.72 ± 21.95 mmHg and a mPAP of 39.46 ± 12.46 mmHg. Exercise capacity, evaluated with the 6MWT, showed a mean distance covered of 276.21 ± 119.31 m, with most patients classified in WHO functional class II-III. Finally, the average number of medications taken was 1.25 ± 0.98, and the reported mortality rate was 47%, with a mean survival of 43.03 ± 22.25 months. By dividing the patients into three clusters, it was possible to make comparisons based on different clinical and demographic characteristics (Table 2).

Cluster 1 had a mean age of 68.57 ± 10.54 years, significantly different from Cluster 2’s 71.36 ± 8.32 years and Cluster 3’s 61.11 ± 13.50 years, with Cluster 3 being the youngest (*p* < 0.001). Significant differences were also observed in respiratory function among the clusters, with Cluster 2 exhibiting significantly lower FVC and FEV_1_ compared to the other clusters, particularly FVC 68.01 ± 12.66 (*p* = 0.001) and FEV_1_ 68.12 ± 10.20 (*p* = 0.015).

Furthermore, patients in Cluster 2 had lower DL_CO_ (*p* < 0.001) and DL_CO_/V_A_ values (*p* = 0.005) compared to those in Cluster 1, while Cluster 3 showed intermediate values between the two, with statistically significant differences.

Pulmonary arterial pressure measurements revealed that patients in Clusters 2 and 3 had higher sPAP and mPAP values compared to Cluster 1 (*p* < 0.001), with Cluster 3 showing the highest values in both measurements (sPAP 82.76 ± 20.02 and mPAP 48.72 ± 11.80; *p* < 0.001). Additionally, Cluster 2 had higher PAdx, RVP, and GTP values compared to the other groups (*p* < 0.001).

Regarding functional capacity, Cluster 2 had a significantly lower 6MWT distance (219.09 ± 79.82 m) compared to the other clusters (Cluster 1 = 318.38 ± 130.34 m, Cluster 3 = 275.95 ± 115.40 m; *p* = 0.001), while the number of prescribed medications was significantly higher in Cluster 3 (Cluster 3 = 2.14 ± 0.71, Cluster 1 = 0.71 ± 0.79, Cluster 2 = 1.08 ± 0.81; *p* < 0.001). Finally, the mortality rate was higher in Cluster 2 (75%) compared to Cluster 1 (33%) and Cluster 3 (38%). These mortality differences were confirmed by Kaplan–Meier curves (Figure 3).

A Logistic Regression predictive model identified three risk factors for mortality: age, the number of medications taken for PH, and NYHA class, with an accuracy of 74.4%.

## 4. Discussion

PH is a life-threatening condition associated with increased mortality regardless of the classification and underlying etiology [4,25]. In this regard, it seems strategically important to have optimal, multidisciplinary and integrated management of this condition [26] in order to diagnose it early, treat it appropriately and improve outcomes [27]. There is a strict need for risk stratification algorithms to define the prognosis of these patients and guide diagnostic–therapeutic decisions.

This study aimed to identify homogeneous patient groups (clusters) within the examined population, analyze their mortality rates, and develop a five-year mortality prediction model for patients with PH using ML techniques.

Through clustering, three distinct phenotypes with varying clinical and demographic characteristics were identified.

Cluster 1: predominantly female (57%), with 69% belonging to clinical groups other than Group 3 (according to the 2022 ESC/ERS classification), characterized by relatively favorable clinical parameters, intermediate age (68.57 ± 10.54 years), and better cardiac and respiratory function.

Cluster 2: Composed of 50% women, with the highest average age (71.36 ± 8.32 years). This cluster includes 44% of patients in Group 3, with worse respiratory function and intermediate cardiac function.

Cluster 3: The youngest group (61.11 ± 13.50 years), mostly male (65%), primarily from non-respiratory PH groups (81%), showing intermediate respiratory dysfunction but the poorest cardiac function.

Mortality analysis revealed that Cluster 2 had the lowest survival rates, with a 75% mortality rate, significantly higher than Cluster 1 (33%) and Cluster 3 (38%). These findings suggest two key considerations. First, more severe pulmonary and cardiac impairments appear to be associated with poorer outcomes, highlighting the potential value of an integrated cardiopulmonary assessment in the management of PH. Second, the observed higher mortality among elderly patients with reduced lung function may reflect the combined influence of age-related physiological decline and underlying pulmonary impairment, rather than a direct causal relationship.

The effects of aging on the respiratory system have been widely studied [28] and result from a miscellany of progressive decline in lung function (secondary to altered respiratory mechanics [29]) and loss of total capillary volume.

On the other hand, age-related vascular stiffening has also been shown to involve the pulmonary vascular bed, with age-related increase of systolic pulmonary artery pressure (sPAP) of about 1 mmHg per decade [30].

Decline in spiro-metric data, including FEV_1_ and FVC, predicted was independently associated with an increased risk of death [31].

This is in agreement with our study: Cluster 1, which has the best cardiac and respiratory function values, has the lowest mortality.

Younger patients with severe cardiac dysfunction often show better survival rates than older patients with intermediate cardiac dysfunction (Cluster 3, younger but associated with worse cardiac function, has lower mortality than Cluster 2, consisting of older but intermediate cardiac function patients), probably due to greater physiological resilience and better response to treatments [32].

The study offers important insights into managing PH by identifying distinct patient clusters and developing a predictive model for 5-year mortality. These clusters underscore the heterogeneity within the PH population, highlighting the need for tailored treatment. A ML model identified key predictors of mortality (age, medications taken for PH, NYHA class) showing that older patients in advanced NYHA classes face higher mortality.

Numerous studies have confirmed that older patients have a higher comorbidity and a reduced ability to recover, as well as that NYHA functional class is a strong predictor of mortality; in fact, patients in more advanced NYHA classes (III and IV) show a significantly lower probability of survival than those in classes I and II.

This evidence agrees with data from the REVEAL registry, which identified predictors of 1-year mortality in patients with PH, finding that advanced age was significantly associated with increased mortality, as well as worse NYHA functional class and more treatment medications [33].

Similar evidence was reached by the COMPERA registry, which confirmed that advanced age is associated with increased mortality risk in patients with PAH [34].

This model provides simplicity and accessibility by focusing on just three easily obtainable variables, offering clinicians a practical tool for assessing patient prognosis and tailoring therapeutic strategies.

Although widely used, existing risk models such as REVEAL (Registry to Evaluate Early and Long-Term Pulmonary Arterial Hypertension Disease Management) [35], COMPERA (Comparative, Prospective Registry of Newly Initiated Therapies for Pulmonary Hypertension) [36], SPAHR (The Swedish Pulmonary Arterial Hypertension Register) [17] and FPHR (French PH registry) [37] scores have limitations, including complexity and reliance on invasive parameters. Other risk-stratification tools have been developed from the US REVEAL (i.e., REVEAL 2.0 risk score calculator and REVEAL Lite 2) [36,38].

The ML model developed in this study offers significant advantages in terms of simplicity, accessibility, and consistent performance across diverse patient populations.

This is not the first study to apply AI to PH. Previous research has employed AI to quantify the extent of pulmonary fibrosis on CT scans, showing a strong correlation with increased mortality [39]. AI has also been used to develop algorithms for detecting PH through chest X-ray images [40]. Furthermore, other studies have explored AI models to predict elevated pulmonary pressures using data from echocardiograms [41] or electrocardiograms [42].

The strength of our model lies in its simplicity, as it is based on only a few parameters. Despite this, it provides a valuable method for phenotyping, helping us identify high-risk patient clusters that may benefit from a more aggressive therapeutic approach.

### Limitations

This study presents several limitations that should be acknowledged. First, although the sample size was sufficient for model development, it may still limit the generalizability of the results to broader or more diverse patient populations. Second, the analysis did not account for certain potentially influential variables, such as genetic predispositions, detailed comorbidity profiles, or socioeconomic factors, which could impact mortality outcomes. Third, the retrospective nature of the data may introduce biases related to data completeness and quality. Lastly, while clustering allowed for the identification of patient subgroups, the clusters themselves are data-driven and may not correspond to clinically recognized phenotypes, potentially affecting interpretability and clinical applicability.

## 5. Conclusions and Future Perspectives

This study identified patient clusters and developed a predictive model for 5-year mortality in PH patients, enhancing the understanding of PH progression and informing clinical decision-making. The integration of traditional statistical methods with ML techniques demonstrates the potential to improve diagnostic accuracy and patient outcomes in PH.

Future research could explore several avenues to enhance the findings of this study. Incorporating additional variables, such as genetic data, longitudinal biomarkers, or imaging features, could also improve predictive performance and provide deeper clinical insights. Moreover, integrating temporal dynamics through time-series models or recurrent neural networks might capture disease progression more accurately. Clinically, prospective validation of the models in real-world settings is essential to assess their practical utility.

Another significant challenge in applying machine learning to PH is the limited availability of high-quality, large-scale patient data. As a rare and heterogeneous disease, PH is often underrepresented in standard clinical datasets, and data collection is further hindered by fragmented care, delayed diagnoses, and variability in clinical documentation. These limitations reduce the statistical power of ML models, increase the risk of overfitting, and limit generalizability. Moreover, the scarcity of labeled data, particularly longitudinal data with confirmed outcomes, poses obstacles to building robust predictive models. Collaborative multicenter data-sharing initiatives and standardized registries are essential to overcome these barriers and advance ML research in rare diseases like PH [43].

Finally, combining clustering with clinician-defined phenotypes could help bridge the gap between data-driven stratification and actionable, personalized treatment strategies.

## Figures and Tables

**Figure 1 bioengineering-12-00408-f001:**
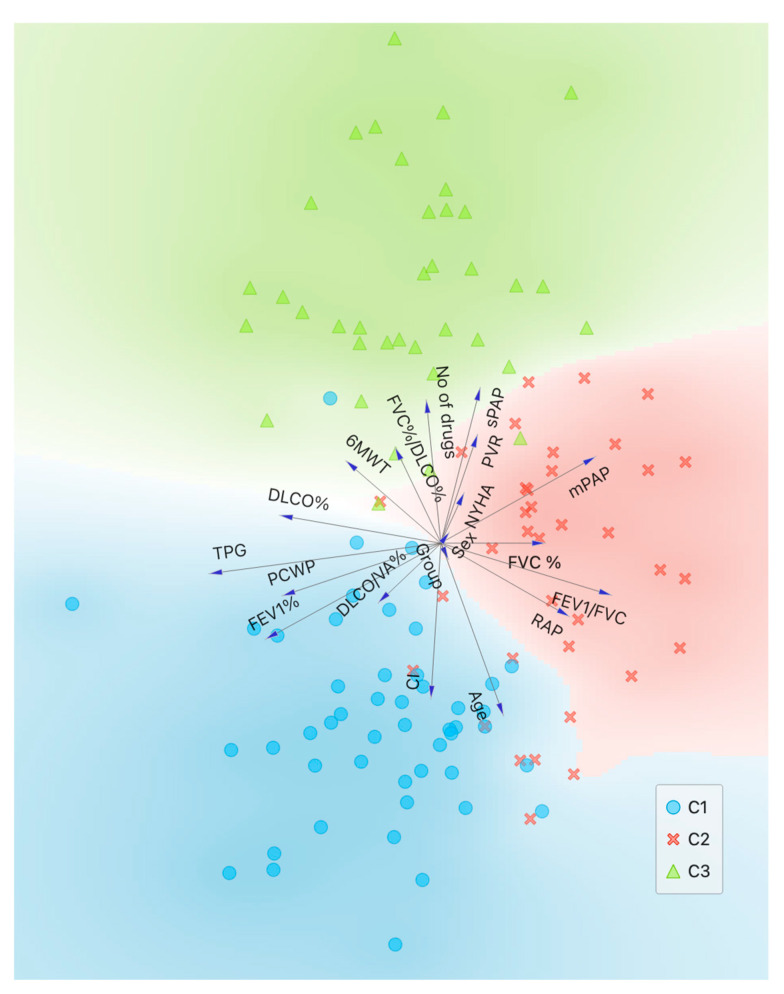
Variables’ contribution to clusters’ identification.

**Figure 2 bioengineering-12-00408-f002:**
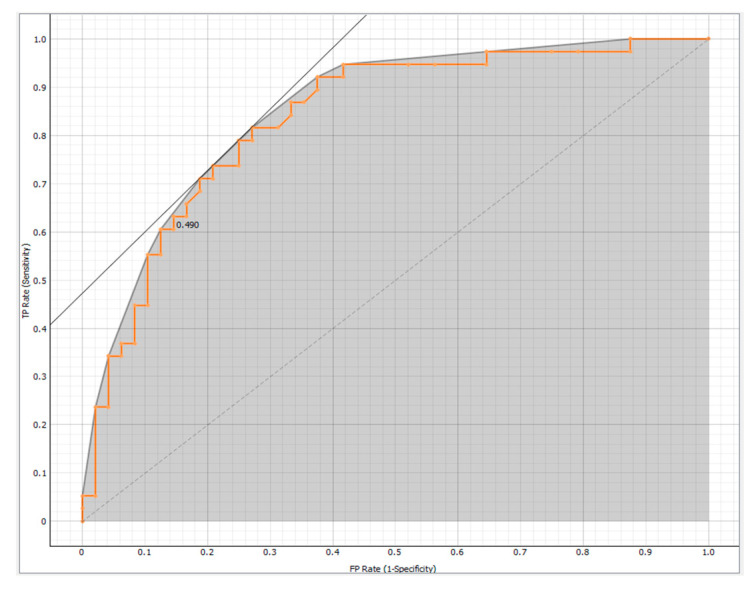
ROC Curve for logistic regression. The Receiver Operating Characteristic (ROC) curve shows the relationship between the true positive rate on the y-axis (TPR, sensitivity) and the false positive rate on the x-axis (FPR, 1-specificity).

**Figure 3 bioengineering-12-00408-f003:**
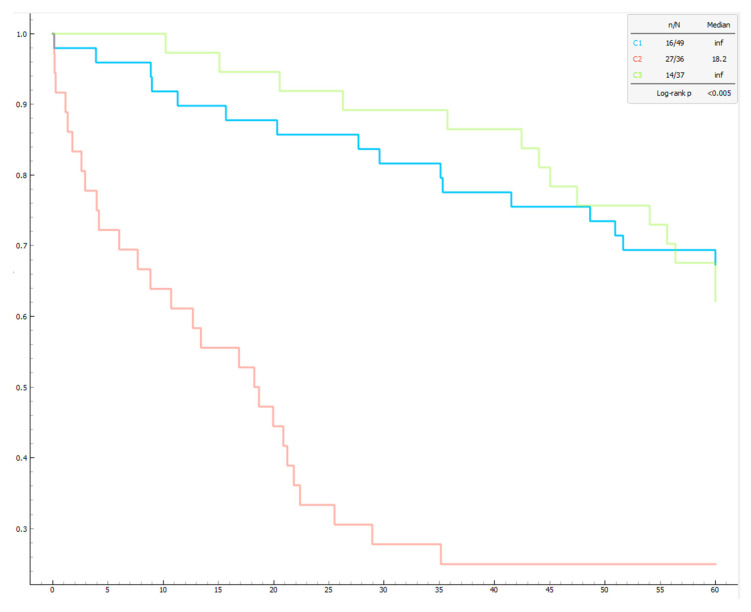
The Kaplan–Meier graph shows significant differences in survival among the three clusters of pulmonary hypertension patients. The log-rank test confirms that the differences in survival between clusters are statistically significant (*p* < 0.005). The x-axis represents 5-year survival in months (from 0 to 60 months), while the y-axis shows the probability of survival (1 indicates 100% survival). The legend indicates the number of deaths out of the total number of patients in each cluster (n/N). The median represents the median survival time. The median is “inf” (infinite) for clusters 1 and 3, meaning that more than half of the patients survived beyond the observation period. C1 (in red) has the lowest survival probability, C2 (in blue) has an intermediate survival probability, and C3 has the highest survival probability.

**Table 1 bioengineering-12-00408-t001:** Population’s characteristics.

Clinical Parameters	N = 122
Age (years)	67.13 ± 11.64
Male (%)	52%
PH Group 1, 2, 4, 5 (%)	69%
PH Group 3 (%)	31%
FVC (%)	77.56 ± 19.10
FEV_1_ (%)	73.12 ± 18.23
FEV_1_/FVC (%)	76.40 ± 13.20
DL_CO_ (%)	48.20 ± 18.92
DL_CO_/V_A_ (%)	74.18 ± 21.95
FVC/DL_CO_ (%)	1.84 ± 0.67
sPAP (mmHg)	68.72 ± 21.95
mPAP (mmHg)	39.46 ± 12.46
PCWP (mmHg)	14.60 ± 6.40
CI (Lmin^−1^mt^−2^)	2.69 ± 1.09
RAP (mmHg)	11.55 ± 6.26
PVR (Wu)	5.83 ± 3.86
TPG (mmHg)	24.83 ± 13.46
6MWT (mt)	276.21 ± 119.31
WHO functional class (n)	2.96 ± 0.73
Number of drugs (n)	1.25 ± 0.98
Exitus (%)	47%
Months of survival (n)	43.03 ± 22.25

Continuous data are expressed as mean ± standard deviation, while categorical as percentage. *Abbreviations:* PH: pulmonary hypertension; FVC: forced vital capacity, FEV_1_: forced expiratory volume in 1 s; DL_CO_: diffusing capacity of the lung for carbon monoxide; V_A_: alveolar volume; sPAP: systolic pulmonary artery pressure; mPAP: mean pulmonary artery pressure; PCWP: pulmonary capillary wedge pressure; CI: cardiac index; RAP: mean right atrial pressure; PVR (WU): pulmonary vascular resistance (Wood units); TPG: transpulmonary gradient; 6MWT: six-minute walking test; WHO: World Health Organization.

**Table 2 bioengineering-12-00408-t002:** Comparison between clusters.

	Cluster 1 N = 49	Cluster 2 N = 36	Cluster 3 N = 37	*p*
Age (years)	68.57 ± 10.54 ^b^	71.36 ± 8.32 ^c^	61.11 ± 13.50 ^b, c^	**<0.001**
Male (%)	43%	50%	65%	0.128
PH Group 1, 2, 4, 5 (%)	69%	56%	81%	0.063
PH Group 3 (%)	31%	44%	19%	0.063
FVC (%)	83.34 ± 19.40 ^a^	68.01 ± 12.66 ^a, c^	79.18 ± 20.71 ^c^	**0.001**
FEV_1_ (%)	78.78 ± 21.54 ^a^	68.12 ± 10.20 ^a^	70.51 ± 17.91	**0.015**
FEV_1_/FVC (%)	76.15 ± 13.29	82.33 ± 9.98 ^c^	70.96 ± 13.70 ^c^	**0.001**
DL_CO_ (%)	58.82 ± 20.91 ^a, b^	38.15 ± 11.50 ^a^	43.91 ± 14.78 ^b^	**<0.001**
DL_CO_/V_A_ (%)	81.49 ± 20.33 ^a^	66.46 ± 18.81^a^	72.03 ± 24.23	**0.005**
FVC/DL_CO_ (%)	1.66 ± 0.55	1.93 ± 0.57	2.00 ± 0.85	**0.044**
sPAP (mmHg)	55.24 ± 16.72 ^a, b^	72.64 ± 19.79 ^a^	82.76 ± 20.02 ^b^	**<0.001**
mPAP (mmHg)	30.22 ± 7.64 ^a, b^	42.50 ± 9.67 ^b, c^	48.72 ± 11.80 ^c^	**<0.001**
PCWP (mmHg)	15.09 ± 6.80	16.35 ± 6.84 ^c^	12.24 ± 4.63 ^c^	**0.017**
CI (Lmin^−1^mt^−2^)	3.23 ± 1.45 ^a, b^	2.33 ± 0.43 ^a^	2.32 ± 0.60 ^b^	**<0.001**
RAP (mmHg)	9.56 ± 4.95 ^a^	15.29 ± 8.06 ^a, c^	10.55 ± 3.95 ^c^	**<0.001**
PVR (Wu)	3.17 ± 1.54 ^a, b^	6.59 ± 2.96 ^a, c^	8.61 ± 4.51 ^b, c^	**<0.001**
TPG (mmHg)	15.32 ± 7.64 ^a, b^	25.95 ± 12.21 ^a, c^	36.32 ± 11.24 ^b, c^	**<0.001**
6MWT (mt)	318.38 ± 130.34 ^a^	219.09 ± 79.82 ^a^	275.95 ± 115.40	**0.001**
WHO functional class (n)	2.61 ± 0.84 ^a, b^	3.18 ± 0.49 ^a^	3.22 ± 0.58 ^b^	**<0.001**
Number of drugs (n)	0.71 ± 0.79 ^b^	1.08 ± 0.81 ^c^	2.14 ± 0.71 ^b, c^	**<0.001**
Exitus (%)	33% ^a^	75% ^a, c^	38% ^c^	**<0.001**
Months of survival (n)	49.58 ± 18.46 ^a^	24.10 ± 22.76 ^a, c^	52.78 ± 13.81 ^c^	**<0.001**

Continuous data are expressed as mean ± standard deviation, while categorical as percentage. Significant differences in Tukey’s post hoc tests are indicated as follows: ^a^ for C1 vs. C2, ^b^ for C1 vs. C3, ^c^ for C2 vs. C3. The significant *p*-value was marked in bold. *Abbreviations:* PH: pulmonary hypertension; FVC: forced vital capacity, FEV_1_: forced expiratory volume in 1 s; DL_CO_: diffusing capacity of the lung for carbon monoxide; V_A_: alveolar volume; sPAP: systolic pulmonary artery pressure; mPAP: mean pulmonary artery pressure; PCWP: pulmonary capillary wedge pressure; CI: cardiac index; RAP: mean right atrial pressure; PVR (Wu): pulmonary vascular resistance (wood units); TPG: transpulmonary gradient; 6MWT: six-minute walking test; WHO: World Health Organization.

## Data Availability

The raw data supporting the conclusions of this article will be made available by the authors on request.

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
