# Peer review of "Utility of Clustering in Mortality Risk Stratification in Pulmonary Hypertension"

_bioengineering, 2025, doi:10.3390/bioengineering12040408_

Round 1

Reviewer 1 Report

Comments and Suggestions for Authors

Thank you for inviting me to review this manuscript on hot topic!

I have several proposals:

  1. please add more limitations of your study
  2. Please analyse in deep how clusterisation can change real life practice and also guidelines in management of pulmonary hypertension
  3. please add future directions for studies
  4. I propose to do visual scheme of phenotypes where can be included all variables which were used for clustering

Author Response

Thank you for inviting me to review this manuscript on hot topic!

I have several proposals:

Comment 1: please add more limitations of your study

Response 1: Dear reviewer, thank you for your comments. We added more limitations to the text.

Comment 2: Please analyse in deep how clusterisation can change real life practice and also guidelines in management of pulmonary hypertension

Response 2: thank your suggestion. We edited the text including more considerations about this aspect of our study.

Comment 3: please add future directions for studies

Response 3: we added future perspectives to the text

Comment 4: I propose to do visual scheme of phenotypes where can be included all variables which were used for clustering

Response 4: We agree you on utility of a visual scheme and we added it (see Figure 1).

Reviewer 2 Report

Comments and Suggestions for Authors Thank you for the opportunity to review this engaging and relevant manuscript. The application of Artificial Intelligence and machine learning in healthcare is an evolving field with significant potential to enhance clinical decision-making, optimize patient outcomes, and improve healthcare efficiency   To further enhance the  impact of your manuscript, I suggest the following revisions: Formatting: 
  • Please ensure that the in-text citations follow a consistent format throughout the manuscript. A uniform reference style will enhance readability and adherence to journal guidelines.
  • In the Introduction section (lines 47–50), there appears to be a formatting issue. Kindly review and correct any inconsistencies
  •  
  • Introduction:
    • The introduction provides a good foundation for the topic; however, it could be expanded to give a more comprehensive overview of how machine learning has been utilized in clinical practice. A broader discussion of its current applications, limitations, and emerging trends would strengthen the context of your study.
    • The paragraph beginning at line 57 would benefit from the inclusion of citations. To support the claims made in this section, please reference relevant literature
      Materials and Methods: 
    • Please provide details regarding informed consent and ethical committee approval to ensure compliance with ethical research standards
      Discussion:
    • The paragraph from lines 216 to 220 is worded in a rather strong manner. I suggest rewording this passage to emphasize that the findings support the observed effect rather than proving causality

Author Response

Thank you for the opportunity to review this engaging and relevant manuscript. The application of Artificial Intelligence and machine learning in healthcare is an evolving field with significant potential to enhance clinical decision-making, optimize patient outcomes, and improve healthcare efficiency   To further enhance the  impact of your manuscript, I suggest the following revisions: Formatting: 

Comment 1: Please ensure that the in-text citations follow a consistent format throughout the manuscript. A uniform reference style will enhance readability and adherence to journal guidelines.

Response 1: thank you for your comments. we edited the references according your suggestions.

Comment 2: In the Introduction section (lines 47–50), there appears to be a formatting issue. Kindly review and correct any inconsistencies

Response 2: the inconsistencies were corrected

Introduction:

Comment 3: The introduction provides a good foundation for the topic; however, it could be expanded to give a more comprehensive overview of how machine learning has been utilized in clinical practice. A broader discussion of its current applications, limitations, and emerging trends would strengthen the context of your study.

Response 3: The text was edited according your suggestions.

Comment 4: The paragraph beginning at line 57 would benefit from the inclusion of citations. To support the claims made in this section, please reference relevant literature

Response 4: thank you for the comment. the reference was added.

  Materials and Methods: 

Comment 5: Please provide details regarding informed consent and ethical committee approval to ensure compliance with ethical research standards

Response 5: The study was conducted according to Helsinki declaration. We added this sentence in the manuscript.

 Discussion:

Comment 6: The paragraph from lines 216 to 220 is worded in a rather strong manner. I suggest rewording this passage to emphasize that the findings support the observed effect rather than proving causality

Response 6: we agree with you and so we edited the text according your suggestion.

Reviewer 3 Report

Comments and Suggestions for Authors

Reviewer Report on “Utility of Clustering in Mortality Risk Stratification in Pulmonary Hypertension” by Tondo P. Tricarico L, et al.

The authors present a well-written manuscript on utilizing clustering and machine learning (ML) approaches to stratify pulmonary hypertension (PH) in diseased patients. Given the significance of ML in healthcare, the application of a smaller dataset with reasonable ML models serves as an excellent starting point to accelerate early phases of drug discovery. This study can be a valuable resource for a broad scientific audience.

While the manuscript is well-structured and informative, I have a few minor suggestions to enhance clarity and completeness:

  • The introduction effectively covers the recent treatment approaches and pathophysiology of PH. However, it would be beneficial to add a few points about current treatment options, including commonly used drugs or other therapeutic interventions. This will help readers understand not only the biology of PH but also the available treatment regimens.
  • It would also be helpful to mention the challenges associated with obtaining patient data for rare diseases like PH and how this affects ML studies.
  • One of the most important aspects of this study is highlighting the limitations of the ML model and sample size. Expanding on these limitations will help readers understand how to carefully apply this technique in their studies and the extent to which it can be effectively used in current models.

Overall Assessment

The manuscript is well-organized and provides an insightful overview of ML for predicting risk stratification in pulmonary hypertension. By incorporating the suggested revisions, the study will become even more relevant and informative for the broader scientific community.

Author Response

The authors present a well-written manuscript on utilizing clustering and machine learning (ML) approaches to stratify pulmonary hypertension (PH) in diseased patients. Given the significance of ML in healthcare, the application of a smaller dataset with reasonable ML models serves as an excellent starting point to accelerate early phases of drug discovery. This study can be a valuable resource for a broad scientific audience.

While the manuscript is well-structured and informative, I have a few minor suggestions to enhance clarity and completeness:

Comment 1: The introduction effectively covers the recent treatment approaches and pathophysiology of PH. However, it would be beneficial to add a few points about current treatment options, including commonly used drugs or other therapeutic interventions. This will help readers understand not only the biology of PH but also the available treatment regimens.

Response 1: Thank you for your valuable contribution. We added more text according to your suggestion

Comment 2: It would also be helpful to mention the challenges associated with obtaining patient data for rare diseases like PH and how this affects ML studies.

Response 2: Thank you for the suggestion. We mentioned the challenges to obtain the necessary data for this study as requested

Comment 3: One of the most important aspects of this study is highlighting the limitations of the ML model and sample size. Expanding on these limitations will help readers understand how to carefully apply this technique in their studies and the extent to which it can be effectively used in current models.

Response 3: Thank you. We edited the section of study limitations and added more text about it.

Overall Assessment

Comment 4: The manuscript is well-organized and provides an insightful overview of ML for predicting risk stratification in pulmonary hypertension. By incorporating the suggested revisions, the study will become even more relevant and informative for the broader scientific community.

Response 4. Thank you for your efforts. We are confident that with your support the manuscript will improve qualitatively 

Round 2

Reviewer 2 Report

Comments and Suggestions for Authors

From my point of view, the article is in line with the aims of the journal and can be published. 
Thank you for the review
I ask you to include the EC code of Foggia, ringa 143, inside the methods. the approval number is mandatory
For the rest, I thank you for this important work 

Author Response

From my point of view, the article is in line with the aims of the journal and can be published. 
Thank you for the review
I ask you to include the EC code of Foggia, ringa 143, inside the methods. the approval number is mandatory
For the rest, I thank you for this important work 

Response: Dear reviewer, thank you for your suggestions to improve the quality of our manuscript. We have added the approval protocol number of our study.